environmental science/ecology/environmental engineering

ecosystem-based adaptation, ecosystem services, synergies, trade-offs, Lake Victoria Basin, nature-based solutions

**Author for correspondence:**
Dorice Agol
e-mail: d.agol@uea.ac.uk

# Ecosystem-based adaptation in Lake Victoria Basin; synergies and trade-offs

Dorice Agol[1], Hannah Reid[2], Florence Crick[2] and Hausner Wendo[3]

[1]School of International Development, University of East Anglia, Norwich Research Park, Norwich NR4 7TJ, UK
[2]International Institute for Environment and Development (IIED), 80-86 Gray's Inn Road, London WC1X 8NH, UK
[3]ADA Consortium, PO Box 3772-00100, Nairobi, Kenya

DA, 0000-0001-5262-8092

Healthy ecosystems such as forests and wetlands have a great potential to support adaptation to climate change and are the foundation of sustainable livelihoods. Ecosystem-based adaptation (EbA) can help to protect and maintain healthy ecosystems providing resilience against the impacts of climate change. This paper explores the role of EbA in reconciling socio-economic development with the conservation and restoration of nature in Lake Victoria Basin, Kenya, East Africa. Using selected ecosystems in the Lake region, the paper identifies key EbA approaches and explores trade-offs and synergies at spatial and temporal scales and between different stakeholders. The research methods used for this study include site visits, key informant interviews, focus group discussions, participatory workshops and literature reviews. An analytical framework is applied to advance the understanding of EbA approaches and how they lead to synergies and trade-offs between ecosystem services provision at spatial and temporal scales and multiple stakeholders. Our results show that EbA approaches such as ecosystem restoration have the potential to generate multiple adaptation benefits as well as synergies and trade-offs occurring at different temporal and spatial scales and affecting various stakeholder groups. Our paper underscores the need to identify EbA trade-offs and synergies and to explore the ways in which they are distributed in space and time and between different stakeholders to design better environmental and development programmes.

## 1. Introduction

Well-managed ecosystems can provide sustainable services that help to build climate-resilient livelihoods (e.g. [1]). With the

triple world crisis of poverty, climate change and environmental degradation, there is a greater level of urgency to manage, conserve and restore global ecosystems to build socio-ecological resilience (e.g. [2]). Fisher *et al.* [3] provide a broad definition of ecosystem services to include different aspects such as ecosystems, structure, processes and their functions as used directly or indirectly, actively or passively by humans to fulfil their health and well-being needs. Ecosystem-based adaptation (EbA)—also known as nature-based solutions (NbS) to climate change adaptation—has become an attractive concept that promotes socio-ecological resilience to climate change impacts (e.g. [4]). EbA is a strategic approach that integrates biodiversity and ecosystem services to help humans respond to the adverse effects of climate change [5–7] and is routinely accompanied by many developmental and environmental co-benefits [8,9].

Interest in EbA (and NbS) has increased over the last decade owing to its promise to reduce vulnerability and increase resilience through ecosystem restoration and management (e.g. [7]). Both theoretical and empirical work has shown that globally, EbA can be effective in building socio-ecological resilience especially in the context of linkages between climate change vulnerabilities and inequality (e.g. [10–12]). EbA considers local concerns and the priorities of poor and marginalized populations which are disproportionately affected by the impacts of climate change and environmental degradation (e.g. [13]). This is highlighted by the Convention on Biological Diversity (CBD) [6] which recognizes the need to involve local and indigenous communities in the implementation of EbA strategies. It has been argued that EbA can easily be integrated with community-based adaptation strategies, promoting local participation in resilience building, particularly for marginalized groups (e.g. [14,15]). The International Union for Conservation of Nature recently published the Global Standards for NbS, which give prominence to stakeholder participation. Criteria four (4) and six (6) of the Standards discuss benefits and trade-offs between different stakeholders, pointing out the importance of their participation in successful NbS [16]. In another example, Woroniecki *et al.* [15] highlight the value of EbA in promoting local empowerment by delivering social benefits to the marginalized socially excluded vulnerable groups.

EbA is gaining prominence in policy development and natural resources management frameworks and is viewed as a key step towards ensuring sustainable development (e.g. [17,18]). The EbA concept is recognized in various international platforms including the CBD, the Paris Agreement and the Intergovernmental Science-Policy Platform on Biodiversity and Ecosystem Services. It is fully embraced by the international donor community [19]. This is because EbA approaches can be cost-effective and economically viable if implemented properly within a strong and enabling institutional, policy and regulatory environment [20]. EbA implementation delivers cross-cutting outcomes for the three inter-related Rio Conventions consisting of the CBD, the United Nations Framework Convention on Climate Change (UNFCCC) and the United Nations Convention to Combat Desertification [21]. Outcomes include improvements in the adaptive capacity of socio-ecological systems and reversing declines in ecosystem services and biodiversity loss. For example, parties to the UNFCCC, including Kenya, are adopting NbS including EbA in their Nationally Determined Contributions (e.g. [22]). The ultimate goal of EbA is to sustain long-term human well-being and the resilience of socio-ecological systems (e.g. [11]). Such approaches can strengthen systems to continue to function and meet long-term human and ecosystem goals despite disturbances. A resilient socio-ecological system is one that improves the health and well-being of humans and the ecosystems which they depend on; this is important for sustainable development.

This case study in Lake Victoria Basin (LVB) builds on scholarly work which demonstrates the potential effectiveness of EbA and its ability to build the resilience of socio-ecological systems by reducing their vulnerabilities and strengthening their adaptive capacities to cope with risks and shocks such as climate change [20]. Assessments of EbA effectiveness often under-report challenges and trade-offs emerging from its application [23]. Without highlighting how EbA outcomes are distributed in space and time, and between different stakeholders, opportunities for learning from mistakes are reduced and understanding about the limitations of EbA and the necessary conditions for its success is limited. Acknowledging and understanding trade-offs, risks and costs of EbA (direct and indirect) can help tackle these challenges. This paper fills this gap by providing a more critical analysis of EbA risks as well as benefits. The objectives of the paper are: (i) explore trade-offs and synergies occurring at different spatial and temporal scales and between different stakeholders owing to the implementation of EbA interventions; (ii) identify what key governance-related factors contribute to the effectiveness of EbA; and (iii) identify how trade-offs and synergies between ecosystem services provision contribute to the long-term sustainability of socio-ecological resilience in LVB.

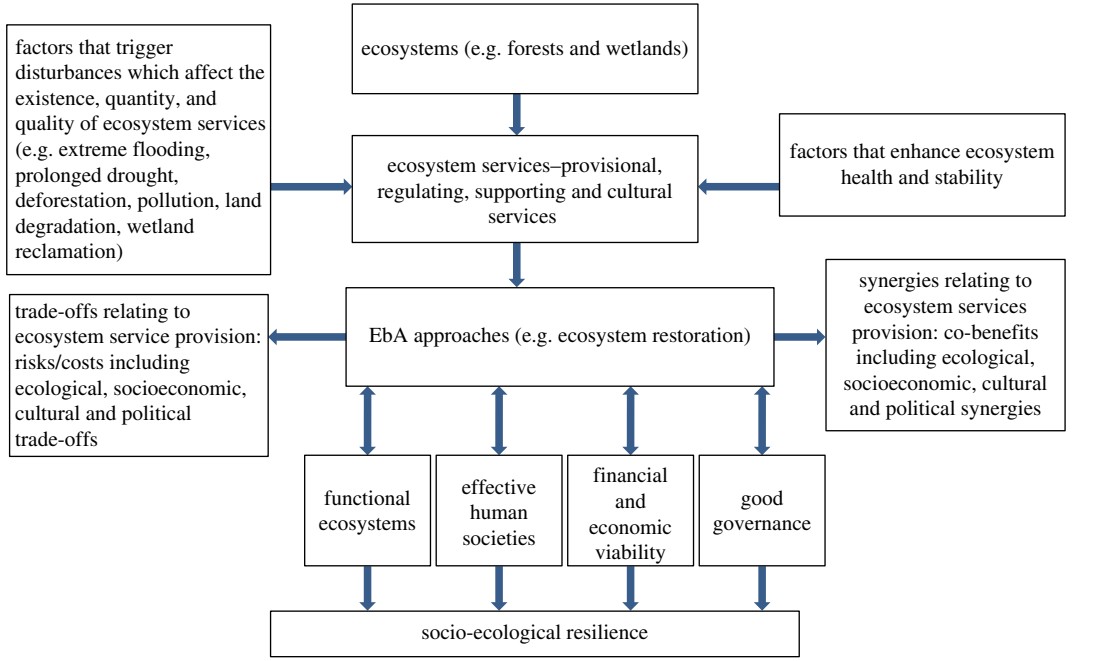

**Figure 1.** Conceptual framework for linking EbA and socio-ecological resilience (LEbASER). The arrows shows connections between ecosystems, the services they provide, factors that enhance or reduce those services, EbA approaches, synergies and trade-offs from the implementation of those approaches, and factors that influence approach effectiveness. This figure shows that implementing EbA requires a good conceptualization of ecosystems, their structures, functions, threats, management approaches and resultant synergies and trade-offs. The framework shows that it is important to explore and understand all aspects of ecosystem services, and their linkages with different aspects of effective EbA which is important for socio-ecological resilience.

# 2. Material and methods

## 2.1. Theoretical framework

The theoretical framework 'Linking EbA and Socio-ecological Resilience (LEbASER)' designed for this study examine synergies and trade-offs emerging from EbA strategies in LVB and the ways in which their outcomes may influence adaptative capacities of ecosystems and humans (figure 1). The framework assumes that EbA techniques and approaches are full of trade-offs and synergies which often coexist in space and time. Trade-offs occur when the maintenance of or improvements in the delivery of one ecosystem service simultaneously inhibits the delivery of another, or when one stakeholder group benefits from ecosystem service-related improvements at the expense of another, either simultaneously or in different dimensions of time or space [24]. In comparison, synergies refer to compatibilities, ripple effects and positive associations whereby the improvements in ecosystem service provision lead to improvements in another, with related indirect benefits accruing to different social groups in time or space (*ibid*). To achieve lasting socio-ecological resilience, a deeper exploration of trade-offs and synergies is required to understand how they influence effective EbA. This framework builds on existing concepts on resilience building, often applied within different socio-ecological systems which explore the inter-relationships between healthy ecosystems and human well-being (e.g. [25,26]).

The current conceptualization of EbA by Reid *et al.* [4] depicts four main fundamental components that underpin its effectiveness. First, EbA strengthens ecological resilience leading to functional ecosystems that can provide effective services. Second, effective EbA builds societal adaptive capacities and resilience and promotes sustained human well-being. Third, EbA can be economically viable, promoting sustainability. Lastly, effective EbA requires good governance with functional institutions, laws and policies [4]. Although this conceptualization of EbA is useful, it does not explicitly show how these four EbA components (i.e. *functional ecosystems; effectiveness for human societies; cost-effectiveness and financial or economic viability and good governance*) are influenced by potential trade-offs and synergies at different spatial and temporal scales and between different stakeholders. This framework fills this gap and proposes a holistic approach that visualizes ecosystems, their structures, functions and threats alongside EbA measures and associated trade-offs and synergies, that all need to be understood in relation to the four

components of EbA that Reid and colleagues have proposed. For example, while an EbA measure can effectively bring societal benefits relating to adaptive capacities and resilience, trade-offs can emerge at the same time generating conflicts between people.

The LEbASER framework is useful for conceptualizing EbA measures as sources of multiple risks, threats and co-benefits, all occurring at spatial and temporal scales and between different stakeholders in the LVB. The framework takes a stepwise approach that shows the complexities of implementing EbA to reconcile socio-economic development with conservation and nature restoration in LVB. It starts with ecosystems such as forests or wetlands and their various functions (e.g. provisional, regulating) and then identifies potential triggers and disturbances (e.g. prolonged drought, extreme flooding). To counter such disturbances requires EbA interventions such as ecosystem restoration; however, such approaches inherently bear costs and benefits. It is therefore necessary to unpack trade-offs and synergies that are associated with any proposed EbA measures to gain a good understanding of the extent to which they fit within three sustainability pillars—social, economic and environmental—and their implications for socio-ecological resilience in the LVB.

## 2.2. Data collection and analysis

This study was conducted in the Kenyan side of LVB between April and June 2019. A qualitative case study approach was used combining literature reviews, semi-structured interviews, focus group discussions (FGDs), a participatory workshop and field observations. Table 1 summarizes all the methods used in data collection.

A content analysis was conducted where data gathered from multiple sources were systematically categorized into themes and interpreted (e.g. [27,28]) with reference to synergies and trade-offs. The content analysis was useful in exploring EbA concepts, trade-offs and synergies, local priorities and expectations on EbA-related activities as well as key opportunities and challenges of implementing EbA measures. For example, we analysed notes from the interviews and FGDs, by identifying common threats to ecosystems and/or ecosystem services benefits that respondents mentioned (electronic supplementary material).

# 3. Background and context

## 3.1. Background of Lake Victoria Basin

LVB covers an area of $68\,800\,km^2$, with a long shoreline of approximately 3500 km [29]. The Lake is a transboundary aquatic ecosystem shared by three East African countries: Tanzania (51%), Uganda (43%) and Kenya (6%) and is an important socio-ecological system, endowed with wetlands, rangelands (drylands), forests, woodlands and farmlands (ibid). The Kenyan side of LVB has a special designation as an economic zone owing to its immense contribution to socio-economic development, including water provision, fisheries, transport and tourism [30]. For example, the fishing industry provides an important source of revenue for local, national and international markets, approximately US$800 million annually [31]. On the Kenyan side of the Lake, there are several administrative boundaries (figure 2) within the Lake's catchment which constitute the Lake Region Economic bloc (LREB) with several rivers passing through them and finally draining into Lake Victoria including: Nzoia, Yala, Nyando, Awach Sondu Miriu and the Mara river (ibid). The main forests on the Kenyan side are; Kakamega, Mau, Elgon and Nandi and Cherangany Hills. These ecosystems support multiple economic activities including agriculture, fisheries and tourism [30]. For example, sugarcane farming is an important economic activity in the Lake Region and accounts for approximately 15% of Kenya's agricultural gross domestic product. Over quarter of a million small-scale farmers are involved with sugarcane farming in the Lake region and derive about 81% of their household income from it (ibid).

## 3.2. Key ecosystem benefits and threats

The ecosystems of the LVB, their services and functions are the foundation of livelihood activities—they are sources of water, food, energy and biodiversity [32,33]. The wetlands of LVB provide water supplies for domestic purposes, irrigation, industrial activities, building and construction materials, medicinal products, biomass and hydropower (e.g. [34]). They also maintain biological biodiversity, regulate the climate, filter pollutants and act as carbon sinks (e.g. [35]). The forested catchment areas of the Lake

**Table 1.** Data collection methods.

| method | function | remarks (e.g. selection criteria, sample sizes) |
| --- | --- | --- |
| literature review | an in-depth review of the literature to identify key ecosystems and their services in the LVB, relevant institutions, regulations and programmes. Various journal articles, policy and briefing notes reviewed to explore EbA concepts, theories and empirical work and key conceptual frameworks on socio-ecological resilience. Key relevant policies and regulations were also reviewed | conducting a literature review helped to identify key informants as well as identify questions for the semi-structured interviews key literature included: journal articles, technical reports, and policy documents on climate change, energy, water, fisheries and agriculture |
| semi-structured interviews | interviews were held with key informants from selected institutions at national, regional (county) and local levels to assess their roles and functions in EbA-related activities as well as their experiences, and the opportunities and challenges they perceive with reference to synergies and trade-offs | respondents were identified from the literature review and using a snowballing approach where key informants helped to identify and recruit potential respondents forty-one (41) people participated in the interviews. Purposeful sampling was conducted based on institutional representation and roles and functions |
| FGDs | a series of open discussions were held with actors and stakeholders of different EbA-related activities in the LVB. The purpose of the FGD was to explore their experiences with EbA activities, and the challenges and opportunities they perceive with reference to synergies and trade-offs | results from semi-structured interviews and literature reviews were used as a guide for FGDs. Five (5) FGDs were held with a total of 24 participants (approx. five people per group) mainly from local communities engaged directly with EbA activities. Participants included men, women and youths |
| field visits and observations | field visits were conducted to observe key ecosystems in LVB, including wetlands, forests, farmlands and rangelands and their various habitats. During field visits, key informants were interviewed and FGDs were held with local communities | literature reviews helped identify specific sites to visit. Conducting semi-structured interviews and FGDs also helped to identify field sites. Four (4) sub-regions (counties) were visited: Kisumu, Siaya, Kakamega and Bomet. These counties were selected because they have major ecosystems e.g. Lake Victoria, Yala Swamp, Kakamega forests and the Mara River rangelands |
| participatory workshop | a workshop was held to validate the study results, and explore participants' views and perceptions on EbA | respondents of the semi-structured interviews and FGDs participated in the validation workshops. More than 50 participants from various national, sub-regional and local level institutions attended |

Basin such as the Mau forest Complex[1], Mount Elgon, Kakamega, Nandi and Cherangany Hills are important sources of the rivers that drain into the Lake [29]. They provide fuelwood, timber, medicinal plants, wild fruit and vegetables, honey, and fodder for livestock as well as services such as

---

[1]The Mau Forest complex is the largest indigenous forest in East Africa.

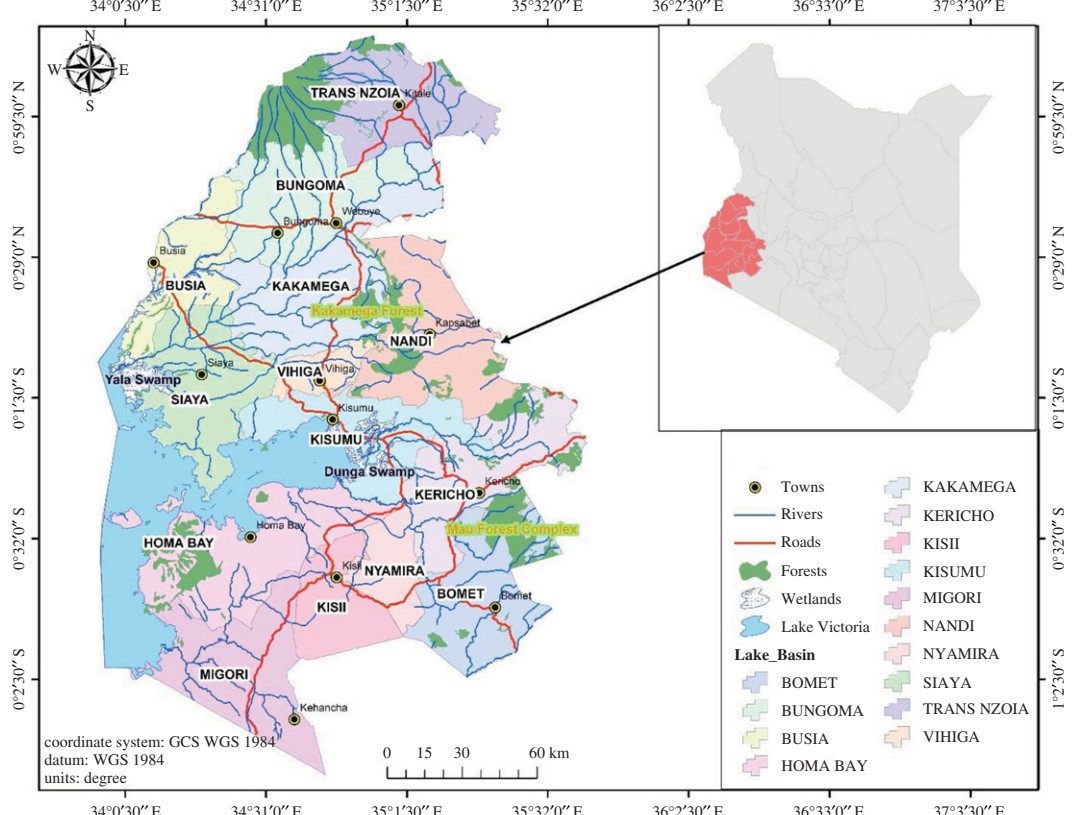

**Figure 2.** Map showing Lake Victoria Basin (creation of the team).

biodiversity protection, climate regulation, flood and soil erosion control, nutrient cycling and soil formation [32]. Many of the natural ecosystems in the catchment of the LVB have important cultural values and are used for education, recreation and tourism activities. The Maasai Mara-Serengeti ecosystem, which is a part of LVB and extends into Tanzania, is globally recognized for its tourism, agricultural, pastoralism and mining activities [36]. The Maasai Mara Game Reserve in Kenya and the Serengeti National Park in Tanzania are popular tourism destinations with rich wildlife and culture[2], attracting millions of visitors from all over the world (*ibid*). Major ecosystems such as Lake Victoria, the Yala Swamp, Lake Kanyaboli, Kakamega forest and river Nyando floodplains are valuable sociocultural sites[3] in the Lake region while the farmland ecosystems support agricultural activities [37,38].

Major threats to LVB's ecosystems and their functions include: climate change, deforestation, soil erosion, pollution and wetland drainage (e.g. [32,33]). Recent and past studies have shown that rising temperatures, prolonged droughts, extreme and erratic rainfall events all have significant socio-economic and ecological impacts on the LVB (e.g. [30,32,33]). Climate variabilities affect major economic activities in the LVB such as fisheries leading to lower yields (e.g. [39]). Unsustainable activities such as deforestation, land quarrying and sand harvesting cause extreme flooding leading to soil erosion across the Lake's catchment especially along riverbanks and in farmlands [32,35]. In forest ecosystems such as the Kakamega, Mau, Nandi and Cheranganyi Hills, deforestation has caused biodiversity losses, decreased species numbers, composition and richness [30]. Pollution and subsequent high nutrient levels in the Lake's wetlands promote algal growth and harmful weeds such as the water hyacinth[4], which has persisted in the Lake since the 1990s disrupting important socio-economic activities such as fishing, transport and recreation (e.g. [35,39]). Many of the wetlands in the LVB have been drained for development purposes and farming activities, causing biodiversity losses. For example, the endemic Sitatunga antelope in Yala Swamp is threatened [40].

---

[2]Indigenous Maasai communities with rich culture and tradition.

[3]E.g. shrines, circumcision, bull and cock fighting, dog markets, traditional dances and pottery etc.

[4]The water hyacinth an invasive weed species which thrives in high temperatures.

**Table 2.** Selected institutions, laws, policies, plans and strategies relevant to EbA in the LREB.

| national, county and local level | institutions, laws, policies, plans, strategies |
|---|---|
| ministries and departments | at national and county government levels including: Climate Change Directorate; environment and natural resources, water and irrigation, agriculture, forestry, planning |
| key agencies (semi-autonomous) | LREB Secretariat., KFS, KWS; |
| | Kenya Forest Research Institute (KFRI), National Environmental Management Authority (NEMA), Kenya Marine & Fisheries Institute (KMFRI) |
| | Kenya Agricultural Research and Livestock Organization (KARLO) Water Resource Authority (WRA) |
| | Department of Resource Surveys and Remote Sensing; National Land Commission and Kenya Meteorological Department (KMD) |
| | Water Resources Users Associations (WRUAs) |
| | Community Forest Associations (CFAs) |
| laws and policies | Climate Change Act (2016) |
| | Water Act (2016) |
| | National Climate Finance Policy (2018) |
| | National Climate Fund Regulations |
| | The County Government Act, 2012 |
| | National Framework Policy on Climate Change (NFPCC) |
| | County Climate Change Bill |
| | Irrigation Act and Policy |
| | National Water Resources Strategy |
| plans and strategies | Vision 2030 |
| | Kenya Nationally Determined Contributions (NDCs) |
| | National Adaptation Plan (NAP) |
| | County Integrated Development Plans (CIDPs) |
| | County Climate Change Fund (CCCF) mechanism |
| | National Climate Change Response Strategy (NCCRS) |
| | Kenya Climate-Smart Agriculture Strategy |
| | Kenya National Biodiversity Strategy and Action Plan (KNBSAP) |
| | Eucalyptus Removal Strategy |

## 3.3. Policy and institutional framework

Vision 2030, which is the national blueprint for development in Kenya, recognizes the need to enhance disaster preparedness in risk-prone areas to improve their capacities to adapt to global climate change [41]. The policy and institutional framework related to EbA in LVB is embedded within various ministries, departments and agencies (table 2). At the national level, climate change adaptation and mitigation activities are supported by ministries such as energy, water, agriculture and natural resources. Several laws and policies on climate change, water, agriculture, forests, land, fisheries and irrigation have been developed and reformed in recognition of the significant impact of climate change on livelihoods and the Kenyan economy. Key policy and regulatory frameworks include the Climate Change Act (2016), Water Act (2016), Environment and Management Act (EMCA 2016), and Forestry Act (2016). For example, the Framework Policy on Climate Change (2016) and Climate Change Act (2016) provide guidance on climate change action through a 5-year short-term plan (currently the National Climate Change Action Plan—NCCAP 2018–2022) and a medium-term adaptation plan

(National Adaptation Plan (NAP) 2015–2030). These documents provide the framework for a low carbon, climate-resilient development pathway through climate adaptation, mitigation and related governance. In particular, the NCCAP places emphasis on climate change adaptation as the main priority for Kenya, as climate change impacts affect all sectors, and the livelihoods and well-being of its citizens [42].

At the county level, the LREB Secretariat leads socio-economic development in the Lake region and protecting the Lake's resources (e.g. through policy-making processes). The Secretariat recognizes that climate change is the biggest threat to the region's socio-economic development [30]. In response, counties within the LREB are keen to reform policies, laws and institutions and mainstream climate change through County Integrated Development Plans (CIDPS) across key sectors such as agriculture, energy, water and sanitation, infrastructure, transport and health [42]. Existing structures—such as the technical LREB Sectoral Committee[5] on Environment, Water and Climate—are responsible for addressing climate change and biodiversity issues. At local levels, key institutions such as Water Resources Users Associations (WRUAs) and Community Forest Associations (CFAs) are also well-established and implement various local initiatives that address climate adaptation and mitigation issues. However, such community-based organizations have limited power and capacities to influence decision-making processes.

Table 2 reveals that the institutional and governance environment for implementing EbA are complex and involve multiple agencies, institutions, laws, policies, plans and strategies. The different actors, and their institutions, policies and regulations, have various roles and responsibilities, some of which are aligned with each other, while others are misaligned. For example, protecting the natural ecosystems of Lake Victoria is a shared responsibility of multiple ministries including environment and natural resources, water and agriculture. This is stipulated in their respective policies and laws such as the Climate Change Act (2016) and Water Act (2016). Devolution adds to this complexity, where the national and the county level governments share responsibilities for implementing climate change adaptation strategies. For example, the recently launched County Climate Change Fund (CCCF), a devolved public fund that ensures that climate finance reaches the most vulnerable populations in each county, needs to be aligned with adaptation mechanisms of the national government.

# 4. Synergies and trade-offs emerging from ecosystem-based adaptation measures in Lake Victoria Basin

Analysis of literature showed that since the early 1990s, several efforts have been made in different parts of the LVB with a common goal of building socio-ecological resilience [30,32,33,35,37,38]. Results from interviews and FGDs revealed that increasingly, there is much awareness of the impacts of climate change and its threats to the Lake Region's economy and ecological integrity. This has led to the implementation of several climate adaptation and mitigation measures in various parts of the Lake Basin in line with the NAP (2015–2030) which demonstrates Kenyan commitment to mainstreaming climate action across all sectors [43]. In this study, we identify and discuss four key EbA-related measures namely: afforestation/reforestation, natural protection of springs, restoration of wetland vegetation and climate-smart agriculture. These measures have been implemented by government agencies, non-governmental organizations (NGOs) and local community groups, and the majority are collaborative initiatives where these actors have different roles and responsibilities in their implementation. Our results show that some of these strategies were not originally designed as NbS but with time, evolved to embrace EbA objectives thus generating multiple co-benefits as well as trade-offs. Table 3 presents different EbA measures and their locations, and outlines resultant synergies and trade-offs occurring at various spatial and temporal scales and between different stakeholders. These trade-offs and synergies are discussed in more detail in §4.1 and §4.2 below.

## 4.1. Synergies and trade-offs at spatial and temporal scales

Our literature review showed that many afforestation and reforestation programmes fall within the larger adaptation and mitigation frameworks in the Lake region and have multiple co-benefits including biodiversity conservation, carbon sequestration and soil erosion prevention (e.g. [40]). During field visits, several afforestation and reforestation activities were observed in the catchment of Kakamega forest. Interviews and FGDs revealed that certain activities were designed as part of EbA measures,

---

[5]Composed of 10 County Executive Committee members of Environment and Natural Resources.

**Table 3.** Summary of synergies and trade-offs from EbA implementation in LVB.

| EbA measure and function | synergies | trade-offs | EbA site (see locations on the map in figure 2) |
|---|---|---|---|
| afforestation and reforestation—to restore forests, increase forest cover and build their adaptive capacities to withstand the impacts of climate change such as storms and floods | increased vegetation cover leading to improved biodiversity (species richness and composition) leading to resilient ecosystems | conflicts between government, local communities and conservation agencies (owing to exclusion from forests) | implemented by KFS, Water Resources Authority, CFAs and WRUAs in Kakamega forest and Mau forests |
| | synergies between government, conservation agencies and local communities | human–wildlife conflicts: wild animals in adjacent restored forests destroy crops and attack humans | |
| | forest restoration upstream improves soil structure and minimizes soil erosion reducing flooding downstream | increased forest cover depletes groundwater resources (e.g. eucalyptus tree species) | |
| | protected upstream forests improve hydrological functions of rivers leading to the continuous flow of water during the dry season | pressure on upstream forests (dry season grazing), rivers and streams in the dry season sometimes leading to conflicts | |
| | synergies between different water users (for household purposes, irrigation and livestock) owing to improved water quantities and quality in rivers | pressure on water sources sometimes leading to conflicts between different water users | |

(Continued.)

**Table 3.** (Continued.)

| EbA measure and function | synergies | trade-offs | EbA site (see locations on the map in figure 2) |
|---|---|---|---|
| spring and riparian protection—to improve water availability and minimize shortages during prolonged drought | spring protection improves water availability during the dry season | intense competition over a few spring water resources available in the dry season when surface water resources are limited and more people to go the protected springs | a spring protected by WRUA in Shinyula Kakamega |
| | spring protection leads to improved groundwater resources (recharge) | increased levels of groundwater use leading to depletion of groundwater resources | |
| | riparian land protection through tree planting protects rivers and improves adaptive capacities of the natural surroundings | certain tree species planted in the riparian zone (eucalyptus) deplete groundwater resources | riparian land protected in Nyangore, River, a tributary of the Mara River, Bomet by the World Wildlife Fund for Nature (WWF) |
| | improved downstream-upstream water user relationships | pressure on water sources sometimes leading to conflicts between different water user groups | |
| wetland restoration (e.g. re-establishment of papyrus)—to improve adaptive capacities of wetlands | improved vegetation cover on the shore leading to better protection of wetlands during storms | conflicts between county governments, conservation agencies and local farmers | Yala Swamp protected and restored by Nature Kenya and Yala Swamp Conservancy Organization in River Yala, and Dunga Swamp on the shore of |
| | wetland vegetation such as papyrus provides refuge for wildlife and encourages eco-tourism | conflicts between county governments, conservation agencies and local farmers | Lake Victoria, Kisumu, protected by a local youth group |
| climate-smart agriculture—to improve resilience in farming systems and livelihoods | agro-forestry improves tree cover and biodiversity within farms and in surrounding environments | organic farming can be expensive for poor farmers with smallholdings | conservation agriculture implemented in the Nyando River and Kakamega forest catchment by local community groups |

for example, in Kakamega, Nandi and Cherengani Hills, Mount Elgon and the Mau forests where afforestation and reforestation programmes are common. Literature shows that within restored areas of these forests, vegetation cover increased, leading to improved biodiversity in terms of species numbers, richness and composition (e.g. [44,45]). Additional afforestation co-benefits noted are improvements in water quality and quantity, and prevention of soil erosion downstream. For example, thick forested areas can withstand extreme flooding events and prevent soil erosion downstream. As many of the forests are important sources of the rivers that drain into Lake Victoria (e.g. the Mara, Yala, Nyando and Nzoia) their restoration can improve the hydrological functions of these rivers and enhance water quality and quantity throughout the year (e.g. [36]).

However, during FGDs with local community groups living in areas adjacent to Kakamega forest, it was revealed that forest restoration activities have caused conflicts in the past, between conservation agencies and local communities. Field observations revealed that although access by local communities is permitted in some parts of the Kakamega Forest to engage with livelihoods activities (e.g. crop cultivation, water and firewood collection) large areas of the forest are protected. Within these, human activities are not permitted, and, in the past, such exclusion has led to conflicts between local communities and conservation organizations.

In Kakamega, a community group protected Sinyula Springs, using earth material and planting vegetation around it to improve its water quality and quality. During FGDs, the group indicated that the Springs provide water all year round which is beneficial in the dry season. Further discussions revealed that well protected springs such as Sinyula facilitate groundwater recharge allowing continuous flow of water throughout the year in both dry and wet seasons. However, we found that during the dry season, the level of water use intensifies as everyone tends to go to the protected springs to fetch water, leading to competition and conflicts between water users.

In Bomet, thick plantations of eucalyptus trees were observed in the riparian zones along Nyangores River, a tributary of the Mara River. Here, local farmers listed numerous benefits of eucalyptus trees including their fast-growing ability and economic viability as they are good sources of timber for sale. Other benefits of eucalyptus noted that were linked with adaptation were control of flooding and soil erosion and improved biodiversity (birds and insects) in the riparian land. However, local resource users also noted potential risks of growing large numbers of eucalyptus. For example, their fast-growing ability and long-extended root system, which enable them to quickly draw large water quantities, can lead to serious impacts on river flow and consequent negative trade-offs for other water users (e.g. [46]). This trade-off has led to a widespread campaign across Kenya, including in the Lake region, to remove the trees, especially those grown near water sources. However, some farmers are unwilling to remove the trees from their farms, and this has led to conflicts between local people and government authorities.

Another EbA measure observed was wetland restoration in the Yala and Dunga Swamps in Siaya and Kisumu counties, respectively. In recent years, the level of effort to involve local communities in conservation activities has increased. Subsequently, EbA measures such as planting vegetation to protect wetlands were found to be commonly implemented by local conservation groups in partnerships with GOs and NGOs. Our findings revealed that improved papyrus vegetation in these wetlands has led to extra co-benefits such as improved water quality and quantity and pollution control leading to ecological resilience within these systems (e.g. [47]).

Past studies have shown that small-scale farmers across the Lake region are disproportionally affected by the impacts of climate change (e.g. [48,49]). Discussion with farmers in the lower Nyando River revealed that crops yields tend to be low owing to high temperatures during prolonged drought. This problem is exacerbated by rampant soil erosion caused by extreme flooding events. Climate-smart agriculture is gaining popularity in these areas as a response. Farm visits showed that farmers have adopted different climate-smart agricultural practices including establishing natural hedgerows, agro-forestry, drought-resistance crops, organic farming, construction of natural ponds (earth dams and water pans) and setting aside small conservation areas. Further analysis showed that more attention is shifting towards adaptation even though some of these measures were not originally designed as EbA. Field observations showed that certain farms have well-established fruit trees such as mangoes, papaya, passion, planted alongside vegetables, maize and pulses. Farm owners stated that fruit trees provide multiple services including provision of food and wildlife habitats, soil fertility enhancement, insect pollination and alternative income from the sale of fruits.

Discussions with local farmers along the River Nyando and parts of Kakamega forest catchment revealed that interest in organic farming is growing owing to its potential to increase sustainable crop yields. In farms where crop yields have improved because of such practices, farmers said that they

were motivated to construct natural ponds (earth dams and water pans). Several earth dams and water pans were observed which collect large quantities of water during the rainy season. Well-managed structures were said to store water for several weeks, providing supplies during prolonged dry seasons, co-benefitting humans, livestock and wildlife such as migratory birds. Further results revealed that some farmers in the Yala and Nyando River catchments were cultivating horticultural crops throughout the year using water from the earth dams. These farmlands also maintained biodiversity in all seasons, providing refuge for insects and other plants. Aquatic species thrive on some farms with permanent ponds. Although those promoting climate-smart agriculture tend to convince farmers about its co-benefits, our study showed that practices such as organic farming remain a challenge for many smallholders because they require long-term investment to achieve sustainable yields. They are also not favourable in the context of smallholders with very small pieces of land where agricultural intensification is the preferred food security strategy.

## 4.2. Synergies and trade-offs between different stakeholders

Our findings showed that the EbA measures such as afforestation and reforestation activities implemented in Kakamega were collaborative initiatives between the Kenya Forest Service (KFS), Kenya Wildlife Service (KWS) and local community groups (e.g. WRUAs and CFAs). Interviews and FGDs revealed that such measures protect and conserve biodiversity and simultaneously strengthen stakeholder synergies. This is because local people are allowed access to certain areas in the forest to practise agro-forestry. Discussions with farmers who have plots within the forest revealed that by allowing them to grow crops, they can diversify their livelihoods from the sale of crops.

However, our findings revealed that the co-benefits generated from forest restoration are not necessarily distributed equally between stakeholders. We found that forest restoration can generate trade-offs in some areas especially when local communities are excluded from forests and/or when their crops are destroyed by wild animals living in the forest. In the Mau Forest for example, findings revealed that conflicts persist between conservationists and local indigenous communities who depend on forests for their livelihoods (e.g. firewood, timber, food). Findings further revealed that human-wildlife conflicts tend to be common around areas adjacent to farmlands and human settlements where crop destruction by wild animals (e.g. monkeys, elephants and buffaloes) and bee attacks on humans and livestock are not uncommon. The KWS is responsible for resolving human-wildlife conflicts through a compensation mechanism, but success levels remain low owing to limited resources.

Field observations showed that wetland restoration through re-establishment of natural vegetation such as papyrus in Yala, Siaya and Dunga was benefiting conservation agencies, farmers and eco-tourism industry stakeholders. FGDs revealed that where vegetation is well re-established, there have been improvements in biodiversity and water quantity. In the Yala Swamp for example, during discussions, members of a local conservation group said that they believed that through their conservation efforts, they have restored habitats for special animal species such as the Sitatunga which is endemic in the area and that they have improved the water storage capacities of the wetlands. By doing this, they are strengthening the eco-tourism industry in the area which is good for the local economy. Although not a direct EbA measure, eco-tourism activities such as boat rides, guided tours, shops and restaurants are rapidly becoming important livelihood sources especially for the youth. However, field observations showed that the restored areas within these wetlands although beneficial to conservationists and tour operators, simultaneously attract local farmers who allocate themselves plots within these areas and grow crops such as maize and vegetables. Further analysis showed that restoring the wetland via special conservation areas can trigger livelihood conflicts between conservation agencies, farmers and local fishers especially when access is denied.

Our study found that EbA measures such as protection of water sources using natural materials can minimize conflicts between water users especially during the dry season when intense competition tends to occur. In forests where afforestation programmes have been successful, and trees are well established, certain streams and rivers flow all year round and provide water supplies for humans and livestock. During discussions, it was noted that in the dry season, pastoralists do not go inside the forests in search of water and pasture because there is improved water availability owing to afforestation. Our results show that in some areas where springs, rivers, streams and earth dams were well protected and functional, co-benefits between downstream and upstream users were enhanced (e.g. Shinyula springs in Kakamega and an earth dam in Nyakach, Kisumu). Local water users asserted that water security levels improved, which benefitted households, livestock and small-scale irrigation enterprises. However, where these water sources were not productive, it was noted that conflicts emerged as many households

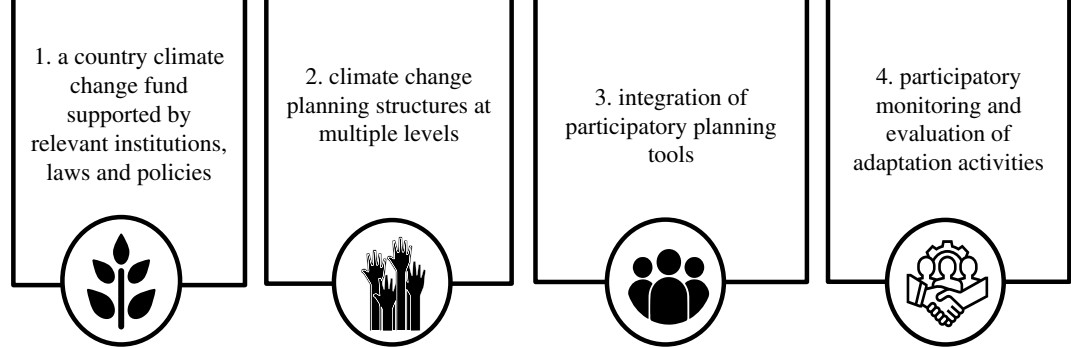

**Figure 3.** Key components of CCCF [52].

scrambled to access the resource. During FGDs, some people expressed their views that only certain households were able to access water from the few available water sources and that priority was given to some water users over others. For example, a local self-help group/organization managing a large tree seedling centre relies on a communal water pan for irrigation, and in the dry season, competition for water tends to intensify leading to conflicts between the group and domestic and livestock water users.

# 5. Governing ecosystem-based adaptation in Lake Victoria Basin

Key informants during the interviews asserted that effective EbA requires an enabling environment with well-established multi-level management and governance structures. This assertion is supported by scholarly literature that shows that effective EbA requires functional institutions, and good policies and legal frameworks that can support adaptive capacities of humans and ecosystems at all levels (e.g. [9,50]). As shown in table 2 above, there are well-established institutions, laws, policies and strategies that support EbA implementation in the Lake Victoria region, at different levels.

## 5.1. Ecosystem-based adaptation governance at national and county levels

At the national level, policy and regulatory frameworks such as the NAP, Climate Change Act (2016), the National Framework Policy on Climate change and the National Climate change Response Strategy and Climate-Smart Agriculture Strategy are examples of key instruments which provide strategic directions for climate adaptation action. At the county level, the CIDPs are among the key policy instruments used for mainstreaming climate change action into development processes.

Certain counties within the LREB have embraced the CCCF mechanism and its core principles of inclusivity, community-driven bottom-up, flexible approaches that focus on public goods investments [51,52]. The CCCF mechanism integrates local, informal (customary) arrangements with formal planning and budgeting processes, to enhance participatory and inclusive governance for sustainable and climate-resilient livelihoods [53]. There are already efforts to mainstream EbA-related activities into county planning, and it is anticipated that the CCCF mechanism being implemented by counties will support EbA activities such as improved management of grazing areas and tree planting at different levels and scales in a more coordinated manner (e.g. [54]). However, our results showed that a key challenge will be to ensure that CCCF funds are properly aligned with EbA priorities across spatial and temporal zones, and between different stakeholder groups, to foster sustainability in the Lake region. During a participatory workshop, stakeholders asserted that for CCCF to be efficient, it would require a strong political will from the county leadership (figure 3).

## 5.2. Ecosystem-based adaptation governance at local levels

Our study shows that there are well-established structures at local levels, many of which govern the implementation of EbA-related activities across the Lake region. Local institutions, such as CFAs, WRUAs, women and youth groups and farmers associations, were found to be important platforms for implementing EbA measures. The Water Act (2016) and the Forest Act (2016) stipulate the formation of CFAs and WRUAs to facilitate local action in forest and water resources conservation, respectively. Our study showed that CFAs are leading community-based afforestation initiatives while

WRUA members are engaged with the protection of water sources (e.g. springs, streams and rivers, water pans and earth dams) by planting trees around them (e.g. in the Mara, Yala and Nyando Rivers).

One of the principal functions of these local institutions is to reconcile conservation with livelihoods needs. For example, illegal encroachment into forests by local communities persists and conflicts between forest managers and local communities are not uncommon (e.g. in Kakamega and Mau forests). To minimize these conflicts, GOs and NGOs are supporting the CFAs, WRUAs and other community groups to engage with income-generating activities (IGAs) such as small businesses, crop production, dairy goat and poultry keeping. Though these are not direct EbA measures, these IGAs help to minimize over-reliance on natural resources. For example, through IGA programmes in Kakamega Forest, members of CFAs were found to engage with alternative livelihood activities to minimize forest destruction. To incentivize communities to engage with afforestation, CFA members are allowed access to special areas within the forest to practise agro-forestry, where permit holders are allocated small plots to grow crops alongside trees. This arrangement enables local people to earn a livelihood while engaging with conservation activities. The key condition is that a permit holder must plant trees. However, during FGDs, it was noted that certain CFA members who were permit holders (of the plots) were not entirely interested in conservation. Rather than being directly involved with agro-forestry practices within their allocated plots, certain individuals have sub-let them to a third party. Furthermore, certain individuals cut down the trees that they had planted within their temporarily owned plots when their permits expired. Further scrutiny revealed that immediate livelihood needs such as food production were more important than long-term investment in conservation activities such as planting trees. These individuals were less interested in conserving public goods such as the forests.

Findings revealed that success in governing the ecosystem services by local institutions such as the CFAs and WRUAS, depends on multiple factors including: their membership, leadership, levels of commitment, power and capacities, property rights, and sustainability and strength of coordination. For example, in Nyangores, a tributary of the Mara River, a WRUA with a membership of more than 600 individuals was found to have strong leadership and a high level of commitment to riverbank protection (e.g. tree planting). However, like many other WRUAs, many of its activities are donor-funded. During discussions with representatives of WRUAs, CFAs and other community groups, it was noted that local expectations of donor-driven projects tend to be high and that many often expect that assistance would continue over a long period of time. However, donor-funded programmes rarely last beyond 6 years and when they exit, the momentum for communal activities tends to reduce. Additional findings showed that limited technical and financial capacities of these locally based institutions led to poor management of EbA-related activities such as protection of springs and earth dams. For example, where water resources infrastructure was poorly implemented (e.g. with poor workmanship), operated or managed, they failed to store sufficient water quantities for longer periods. Poor local capacities were also found to limit coordination of EbA-related activities in the Lake region. Field visits to selected communal projects (e.g. afforestation projects) and discussions revealed that although some efforts have been made to facilitate upstream and downstream coordination, (e.g. umbrella WRUAs or CFAs), financial and technical capabilities are limited. To support local communities, many EbA measures tend to be collaborative initiatives between the government, NGOs and the local community groups. However, we found that the power relationships were often asymmetrical where the GOs/NGOs have more power and authority where they still largely influence the implementation of many EbA activities.

Furthermore, our findings showed that the level of local compliance with conservation policies that support EbA across different parts of the Lake region was low. For example, the Water Act (2016) stipulates that to protect riverbanks, riparian landowners must set aside an area within their farms for conservation activities. However, discussions with riparian landowners revealed that many farmers with small plots of land were not willing to set aside their land for conservation activities as crop cultivation remain their priority. Further results showed that in some parts of the Lake region, land use policies have been reformed to address challenges related to compliance and conflicting land use (e.g. under the Sustainable Land Management programme/policy in Kakamega and Yala). In Yala ecosystem, for example, the new land policy promotes the integration of conservation with socio-economic benefits. However, weak enforcement of such policies remains a huge challenge for EbA measures.

# 6. Discussion

Numerous studies have highlighted the value of effective EbA in delivering social change and benefits to marginalized, socially excluded vulnerable groups, and empowering them to adapt to change and

increase their resilience (e.g. [9,12,15,20,55–57]). A pre-requisite for affective EbA is the need for a deeper understanding of the trade-offs and synergies that are often inherent in many adaptation programmes (e.g. [20]). This study has identified key trade-offs and synergies which are linked to selected EbA measures in the Lake Victoria region at spatial and temporal scales and between different stakeholders.

EbA is still largely advocated as the ultimate solution for building local adaptive capacities and resilience, but in this study, we find that effective EbA is context-dependent and we argue that it is not useful to conceptualize it as a 'one size fits all' solution. Under certain circumstances, EbA can create opportunities to enable natural systems and people to adapt effectively and respond to climate risks with the right institutional support. However, under the same circumstances, an EbA measure can generate conflicts. Thus, an adaptation response programme designed with the aim of building local resilience may be maladaptive for certain groups (e.g. women) within the same area. While EbA can reduce disasters and risks, its implementation can generate complex trade-offs and eventual maladaptation for some stakeholder groups, or over time and space. Risks such as climate change or soil erosion can affect households differently and it is important to identify the people who are most at risk without necessarily making presumptions that everyone who is vulnerable will benefit from EbA implementation (e.g. [58]). A notable example in Lake Victoria, with regard to temporal trade-off is spring protection and riparian land protection which improves water availability in the dry season but that leads to intensity of water use in protected springs where everyone goes to the springs to fetch water. We argue that EbA is not always a 'win-win' situation and that there is danger in bundling all potential EbA beneficiaries together and labelling them as 'vulnerable communities' whose adaptive capacities need to be built.

Our study supports the notion that EbA requires multi-level and collaborative forms of governance involving multiple stakeholders in planning and decision-making (e.g. [59]). Certainly there is an enabling environment for implementing EbA programmes in Lake Victoria, through established GOs, NGOs and community-based organizations (e.g. the LREB secretariat, the KFS, Nature Kenya, WRUAs, CFAs), policies and strategies. To successfully operationalize the concept of multi-level and collaborative governance requires understanding risks and benefits associated with different EbA approaches at all levels. A multi-level governance structure requires a deeper understanding of the institutional structures and modalities, their power, capacities (human and financial), culture and behaviour to identify opportunities and constraints for effective implementation of EbA activities. We have shown that existing community-based institutions such as WRUAs and CFAs play important roles in facilitating local engagements with climate adaptation activities. However, these community-based organizations lack the power and sufficient institutional capacities to implement effective EbA, and to respond to climate change and changing ecosystem dynamics, including the ability to identify climate change risks. Current scholarly work suggests that socio-ecological resilience is strongly connected to and influenced by power relationships which are embedded within wider social, economic, political and cultural structures (e.g. [60]). We argue that it is important to acknowledge power differences between the agencies, the inherent misalignment of their roles and fragmentation of policies. Evidence shows that many adaptation programmes involve multiple agencies with different power relationships [61,62]. In Lake Victoria, power is seldom balanced owing to misalignments of roles and responsibilities as well as policies and regulations. Some actors from the NGOs and GOs remain powerful and largely have influence and control over many adaptation programmes.

Effective EbA requires an integrated landscape/watershed level approach, which involves coordinating a myriad of adaptive activities at different levels and scales and across boundaries. However, ecological boundaries between the various ecosystems in the LVB do not necessarily match administrative borders. With a unique landscape featuring multiple geophysical and administrative boundaries, efforts to coordinate EbA activities across the LBV can be very challenging. Many of the current EbA-related activities are implemented separately at multiple levels and scales, under different land tenure systems and supported under different externally funded programmes. Thus, an integrated planning approach at the watershed or catchment level remains a huge challenge in the Lake Victoria region particularly for land use planning. Many conservation agencies and development practioners in the Lake region are still struggling to conceptualize how to manage its transboundary ecosystems effectively. This problem is universal partly owing to limited understanding of landscape approaches to ecosystem management (e.g. [63]). Taking a landscape approach for effective EbA will therefore require strong coordination mechanisms to align all the different efforts and programmes across the LVB. An important perquisite for strong coordination is mainstreaming EbA in planning processes to foster sustainability (e.g. [50]). The CCCF mechanism offers opportunities to do this; however, it will require a strong political will particularly from the county governments.

# 7. Conclusion

This case study of LVB reaffirms that EbA approaches are characterized by trade-offs (risks and costs) and/or synergies (co-benefits), many of which are unequally distributed (leading to winners and losers). Most EbA approaches in Lake Victoria are considered beneficial because they promote resilience, but less attention has been paid to their associated risks, i.e. trade-offs. Using the LEbASER framework, this study has filled this gap by deepening understanding of trade-offs as well as synergies that occur as a result of implementing EbA measures in the Lake region.

EbA is often argued to bring benefits, and yet adaptation programmes can also cause unintended negative consequences. Such consequences need to be identified as early as possible during project planning and implementation to ameliorate potential risks and costs where possible. Many agencies in the Lake Victoria region promote EbA as an ideal solution to adaptation and resilience, but few foresee it as a potential source of incompatibilities and conflicts. Conservation agencies in particular, still largely promote EbA in the LVB as the perfect approach to climate resilience. EbA trade-offs and risks are not yet sufficiently understood and articulated at different temporal and spatial scales and for different stakeholders in the Lake region. Understanding the limits and constraints to effective EbA is important, and documenting lessons learned from case studies—including what works and what does not work—can help with the design and implementation of effective EbA programmes suited to the local context.

Ethics. This research was done according to ethical clearance procedures of the International Institute for Environment and Development (IIED). All respondents signed informed consents before participation. All names were anonymized, and data protection and confidentiality were observed.

Data accessibility. The datasets supporting this article have been uploaded as part of the electronic supplementary material. Document name: Agol *et al._Supplementary_Material.docx.

Authors' contributions. D.A., F.C. and H.R. contributed to the design of data collection tools. D.A. collected data in the field, did the analysis and drafted the manuscript. H.R., F.C. and H.W. gave critical comments on the manuscript and shaped its intellectual content.

Competing interests. We declare we have no competing interests.

Funding. This research project was funded by the Department of International Development (DFID), United Kingdom. It is under the Deepening Democracy Programme GB -1-204437.

Acknowledgements. We thank all those who participated in the research particularly representatives of governmental and non-governmental organizations and local community groups.

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
