## [Peer Review File · Royal Society Open Science]

Review History

RSOS-201847.R0 (Original submission)

Review form: Reviewer 1 (David Harper)

Is the manuscript scientifically sound in its present form?

No

Are the interpretations and conclusions justified by the results?

Yes

Is the language acceptable?

Yes

Do you have any ethical concerns with this paper?

No

Have you any concerns about statistical analyses in this paper?

Yes

Recommendation?

Major revision is needed (please make suggestions in comments)

Comments to the Author(s)

Thank you for your manuscript. I think this is an excellent study which will make an important contribution to the advancement of this trans-disciplinary approach. I do not feel however, that it is written in a way which will give readers will be able to give most readers clear 'take-home' messages.

I have recommended to the editors that the manuscript is re-written in a conventional way Introduction->Methods->Results->Discussion. As it is currently written, the only hard results are the examples within brackets of more general statements (e.g. page 12, lines 36-36). I do not think you should assume that the majority of readers will read the supplementary material alongside the manuscript - that should really only be available for specialists. I think the manuscript should be self-contained and thus contain quantitative, analysed, results.

Review form: Reviewer 2

Is the manuscript scientifically sound in its present form?

Yes

Are the interpretations and conclusions justified by the results?

No

Is the language acceptable?

Yes

Do you have any ethical concerns with this paper?

No

Have you any concerns about statistical analyses in this paper?

No

Recommendation?

Major revision is needed (please make suggestions in comments)

Comments to the Author(s)

This is a timely and useful paper which holds a lot of potential but needs to be revised to fully bring out its contribution. Please see the comments in the attached (Appendix A).

Decision letter (RSOS-201847.R0)

Dear Dr Agol

The Editors assigned to your paper RSOS-201847 "Ecosystem-based adaptation (EbA) in Lake Victoria Basin: Synergies and trade-off" have now received comments from reviewers and would

like you to revise the paper in accordance with the reviewer comments and any comments from the Editors. Please note this decision does not guarantee eventual acceptance.

Please submit your revised manuscript and required files (see below) no later than 21 days from today's (ie 8th of April 2021) date. Note: the ScholarOne system will 'lock' if submission of the revision is attempted 21 or more days after the deadline. If you do not think you will be able to meet this deadline please contact the editorial office immediately.

on behalf of Dr Agnieszka Latawiec (Associate Editor)
openscience@royalsociety.org

Reviewer comments to Author:
Reviewer: 1

Comments to the Author(s)

Thank you for your manuscript. I think this is an excellent study which will make an important contribution to the advancement of this trans-disciplinary approach. I do not feel however, that it is written in a way which will give readers will be able to give most readers clear 'take-home' messages.

I have recommended to the editors that the manuscript is re-written in a conventional way Introduction->Methods->Results->Discussion. As it is currently written, the only hard results are the examples within brackets of more general statements (e.g. page 12, lines 36-36). I do not think you should assume that the majority of readers will read the supplementary material alongside the manuscript - that should really only be available for specialists. I think the manuscript should be self-contained and thus contain quantitative, analysed, results.

Reviewer: 2

Comments to the Author(s)

This is a timely and useful paper which holds a lot of potential but needs to be revised to fully bring out its contribution. Please see the comments in the attached.

===PREPARING YOUR MANUSCRIPT===

===PREPARING YOUR REVISION IN SCHOLARONE===

<https://royalsociety.org/journals/authors/author-guidelines/#supplementary-material> to include a suitable title and informative caption. An example of appropriate titling and captioning may be found at https://figshare.com/articles/Table_S2_from_Is_there_a_trade-off_between_peak_performance_and_performance_breadth_across_temperatures_for_aerobic_sc_ope_in_teleost_fishes_/3843624.

Author's Response to Decision Letter for (RSOS-201847.R0)

See Appendix B.

Decision letter (RSOS-201847.R1)

Dear Dr Agol,

It is a pleasure to accept your manuscript entitled "Ecosystem-based adaptation (EbA) in Lake Victoria Basin: Synergies and trade-off" in its current form for publication in Royal Society Open Science. The comments of the reviewer(s) who reviewed your manuscript are included at the foot of this letter.

on behalf of Agnieszka Latawiec (Subject Editor)
openscience@royalsociety.org

Appendix A

Ecosystem-based adaptation (EbA) in Lake Victoria Basin: Synergies and trade-off

Review

Overall comment: the focus on trade off and synergies over time and space is a useful and timely topic. The manuscript has a lot of potential and needs to be further refined. Many interesting questions are brought up in section 7 but the analysis falls short of explaining how these questions play out in the LVB case study.

Language needs editing and polishing at times, the writing changes between sections and should be consolidated to 'one voice'.

1. Summary

L40-41: implicit causation which does not seem justified. Unequal distribution is not due to trade off but to other elements (e.g. land tenure rights and access, intra household dynamics and political decisions). The trade off in itself I would argue is a political decision made by one or several stakeholders. The 'elements' (see above) are the ones that entail and mediate the distribution.

- ➔ I would want to have the results and contribution in the summary. What is in the summary at the moment is not precise enough, I would want to know immediately why this paper contributes to the literature. The fact that ecosystems can provide multiple benefits is well known/not contested.

2. Introduction

L1-2 I would argue this challenge occurs everywhere not only in low income countries (LIC). Need to change sentence to reflect that it is the case globally but explain what is specific to this challenge in LIC context.

L4-5 or at least this is what EbA claims to be in the UNEP official narrative

L12-13 examples of these policy development or NR mgmt frameworks? Any overarching international agenda e.g. IPBES agenda or others?

- ➔ the end goal of SE resilience seems to be taken for granted. While the motivation for why it is necessary (due to CC) is there, the why is resilience in the face of climate change a desirable objective is not articulated. It just needs an additional sentence.
- ➔ I miss a short definition of ES. The one by Fisher et al., 2009 would nicely fit your study <https://www.sciencedirect.com/science/article/abs/pii/S0921800908004424> you should find it useful for your paper's approach as well, in particular fig 5. Could help with the trade off beneficiary discussion.

3.1 theoretical framework

L12 disturbance factors seems to be limited to climatic event but should also include human induced interference such as over harvesting, degradation etc

I miss an explanation on what the arrows indicate. For example the arrow between ES box and EbA approach box does not describe the same relationship as the arrow between say ecosystems and ES or between disturbance factors and ES boxes.

Please label all arrows. + the dotted arrows, unclear what 1. They point at (the whole or a sub circle?) and 2. What the dots indicate compared to the blue arrows.

I would strongly advise revising this visual.

- ➔ Generally, the visual needs to be streamlined to be more effective, understandable at a glance.
 - e.g. why the different colours for the circles? If different colours don't indicate differences in categories/types (they all seem to be underpinning factors), I would advise keeping same colours for all sub circles.
 - E.g. disturbance factor box and enhance factor box are both the same 'type/category' these are modifiers of ES quantity and quality in opposite directions, is there a simpler way of representing this?
 - Is the ecosystem box necessary? This is for a specialist audience, so might not be
 - The strength of your study is to focus on trade offs and synergies over time and space, but the framework does not develop it that much, in my opinion, this is what the framework should be about: how to articulate and identify the trade offs and synergies across space and time of EbA so that this can support design/implementation/enforcement – possibly this could be iterative

L54-55 what is systematic about this approach? Please develop

+ the framework does not disaggregate trade-offs and synergies. It articulates perhaps although this is not very much reflected in figure 1

3.2 data

Useful table. You could add an explanation on how the different methods build and feedback on each other, at the moment they seem parallel tracks, while they certainly informed each other.

L14 to 23 does not add anything different to what is in the table, so it can be taken out

L56-57 what is a content analysis?

L58 'to make inferences'? I find this confusing, please rephrase

4. background

L20-21 any indications of the share (relative to each other) and size of each of these economic activities on the Kenyan side (or even by county if available)

4.2 key ecosystem

L27-28 climate variabilities do not have high levels of risks, rephrase please.

L23-40 it would be useful to have an indication of the intensity of these damages. E.g. ha of forest and wetlands lost; level of pollution increase in % etc for each degradation cited.

4.3

L12-13 can you be more explicit on the mandate of LREB, 'facilitate policy processes' is a bit vague

L20 water resource user associations – words need to be swapped

- ➔ This section needs to be refined. It should show the fragmentation and overlap across different ministries, agencies whose mandates relate to different dimensions of EbA (design, management and enforcement for example). i.e. one ministry is overseeing X which related to eba as mandated in Y legislation. Implementation is across xyz agencies which depend on different ministries.

This section should be about showing misalignment or, on the contrary, alignment between different official governance stakeholders.

5. synergies

On Table 3:

- 'no reported trade off' cell in the afforestation and reforestation line: there is one trade off, which is that more forest cover upstream reduces water availability downstream since forest are water users and act as water towers. Of course it means potentially improved baseflow during dry season, but it might mean decreased water availability for downstream water users
- 'intense competition' cell in the spring and riparian protection line: this needs to be somewhat better explained, how is the trade off of riparian protection intense competition over resource when its synergy is improved availability over resource?

Table 3, last column would be very useful as a map locating the ecosystems named. It would help text comprehension for the rest of the paper.

5.1 and 5.2 synergies and trade offs

This section is solely descriptive and a summary of FGDs and interviews. It also needs to be more closely referencing table 3.

I miss an understanding of who decided on the eba measures listed. A discussion on autonomous adaptation and centrally designed and commanded adaptation would be helpful – it would tie back to section 4.3.

- ➔ I recommend linking 5.1 and table 3 more closely and clearly, otherwise this reads as another conceptual framework in table 3 and then summary of FGDs + a discussion is needed of who decided, who implemented and who benefits from these eba measures
- ➔ The same remark applies to 5.2, it is heavily descriptive and ideally would go into a working paper which could be referred to. + It needs to be tied to table 3

6. Governing

L53-54 the need for enabling environment and well established etc is not a conclusion drawn from the results. Or at least the description in section 5 implicit points at it but does not analyse it.

6.1 eba governance at national level

Please add how ecosystems and eba are considered in the CCF mechanism, are they branded as an adaptation/mitigation opportunity? How are they included, if at all.

Why is the CCF not mentioned in section 4.3? perhaps this section should be merged with the reworked 4.3.

6.2

Same title as 6.1? is this a mistake?

L37-38. Low compliance is also due to weak enforcement

7. discussion

The start of the discussion goes over points already brought up earlier in the manuscript.

L3-4 I expected the paper to be precisely about this: who benefits directly, indirectly, who disbenefit. The time dimension announced at the beginning of the paper is not prominent in the results or discussion.

L26-28 precisely, and how does this play out in the LVB case study?

L48-51 again, and how can your articulation in this paper ensure this?

- These points are all very interesting and should be the subject of the actual analysis of the paper.

Appendix B

Response to reviewers' comments

Ecosystem-based Adaptation (EbA) in Lake Victoria Basin; Synergies and Trade-offs

Dorice Agol¹, Hannah Reid², Florence Crick² and Hausner Wendo³

¹*School of International Development, University of East Anglia, Norwich Research Park, Norwich NR4 7TJ, United Kingdom*

²*International Institute for Environment and Development (IIED), 80-86 Gray's Inn Road, WC1X 8NH, United Kingdom*

³*ADA Consortium, P.O. Box 3772-00100, Nairobi, Kenya*

Reviewer: 1

Comments to the Author(s)

Thank you for your manuscript. I think this is an excellent study which will make an important contribution to the advancement of this trans-disciplinary approach. I do not feel however, that it is written in a way which will give readers will be able to give most readers clear 'take-home' messages.

I have recommended to the editors that the manuscript is re-written in a conventional way Introduction->Methods->Results->Discussion.

This study used qualitative research methods as outlined in Table 1. Results are spread out in multiple sections starting from section 4 which shows results mainly from literature review. Then we have sections 5 and 6 covering mostly results from interviews, focus group discussions and observations. We took the approach of presenting results in these multiple sections and discussing them at the same time because structurally this made more sense in the context of this paper than trying to separate qualitative observations and the implications of these observations out into different sections. The discussion section then highlights the key findings.

As it is currently written, the only hard results are the examples within brackets of more general statements (e.g. page 12, lines 36-36).

This study took a qualitative approach and we have presented specific results in various sections for example as shown in Table 3

I do not think you should assume that the majority of readers will read the supplementary material alongside the manuscript - that should really only be available for specialists. I think the manuscript should be self-contained and thus contain quantitative, analysed, results.

The study used qualitative research methods. We do not have quantitative data and have provided the qualitative data as the supplementary material. We feel that the paper is self-contained and can be read and understood without reference to any supplementary material.

Reviewer: 2

Comments to the Author(s)

This is a timely and useful paper which holds a lot of potential but needs to be revised to fully bring out its contribution. Please see the comments in the attached.

Ecosystem-based adaptation (EbA) in Lake Victoria Basin: Synergies and trade-off Review

Overall comment: the focus on trade off and synergies over time and space is a useful and timely topic. Agree

The manuscript has a lot of potential and needs to be further refined. Many interesting questions are brought up in section 7 but the analysis falls short of explaining how these questions play out in the LVB case study.

Language needs editing and polishing at times, the writing changes between sections and should be consolidated to 'one voice'.

Agree and we have done that.

1. Summary

L40-41: implicit causation which does not seem justified. Unequal distribution is not due to trade off but to other elements (e.g. land tenure rights and access, intra household dynamics and political decisions). The trade off in itself I would argue is a political decision made by one or several stakeholders. The 'elements' (see above) are the ones that entail and mediate the distribution.

🕒 I would want to have the results and contribution in the summary. What is in the summary at the moment is not precise enough, I would want to know immediately why this paper contributes to the literature. The fact that ecosystems can provide multiple benefits is well known/not contested.

We have replaced this:

'Findings show that restoration and management of ecosystems such as forests and wetlands can generate multiple adaptation benefits. However, these benefits are not necessarily distributed equally in space time and between humans because trade-offs tend to occur. Results demonstrate the complex nature EbA in reconciling socioeconomic development with ecosystem conservation and restoration'

With this:

Our results show that EbA approaches such as ecosystem restoration have the potential to generate multiple adaptation benefits which bear with them synergies and trade-offs occurring at different temporal and spatial scales and affecting various stakeholder groups. Our paper underscores the need to identify EbA trade-offs and synergies and to explore the ways in which they are distributed in space and time and between different stakeholders to design better environmental and development programmes.

2. Introduction

L1-2 I would argue this challenge occurs everywhere not only in low income countries (LIC). Need to change sentence to reflect that it is the case globally but explain what is specific to this challenge in LIC context.

Agree that climate change vulnerabilities are linked directly with inequality and this is a global phenomenon. We have changed the sentence to reflect this.

L4-5 or at least this is what EbA claims to be in the UNEP official narrative

Yes certainly. This claim is well articulated in a series of briefing notes by UNEP-WCMC and UNEP which we have now referenced. We have also referred to similar notions by IUCN, including in their 2020 Global Standards for Nature-Based Solutions (NbS)

L12-13 examples of these policy development or NR mgmt. frameworks? Any overarching international agenda e.g. IPBES agenda or others?

We have added the sentences below to show that EbA is recognized in the international agendas.

EbA concept is universally recognized in various international platforms including the Convention on Biological Diversity (CBD), the Paris Agreement and the Intergovernmental Science-Policy Platform on Biodiversity and Ecosystem Services (IPBES) and fully embraced by international donor community (UNEP-WCMC and UNEP, 2019)

The concept delivers cross-cutting outcomes for the three inter-related Rio Conventions consisting of the Convention on Biological Diversity (CBD), the United Nations Framework Convention on Climate Change (UNFCCC) and the United Nations Convention to Combat Desertification (UNCCD) (UNCCD, 2017); these include on improving adaptive capacity of socio-ecological systems, reversing decline in ecosystem services and reversing biodiversity loss. For example, parties to the UNFCCC, including Kenya, are adopting NbS including EbA in their Nationally Determined Contributions (e.g. Republic of Kenya, 202

☉ the end goal of SE resilience seems to be taken for granted. While the motivation for why it is necessary (due to CC) is there, the why is resilience in the face of climate change a desirable objective is not articulated. It just needs an additional sentence.

Absolutely! We have added a sentence here as follows;

This can strengthen systems to continue to function and meet long-term human and ecosystem goals despite disturbances. Thus a resilient social-ecological system is one that improves the health and well-being of humans and the ecosystems which they depend on and this is important for sustainable development.

☉ I miss a short definition of ES. The one by Fisher et al., 2009 would nicely fit your study

<https://www.sciencedirect.com/science/article/abs/pii/S0921800908004424>

you should find it useful for your paper's approach as well, in particular fig 5. Could help with the trade-off beneficiary discussion.

Thank you. This is a useful definition which takes into account all the different definitions of ES which have included:

Fisher et. al (2009) provides a broad definition of ecosystem services to include different aspects such as ecosystems, structure, processes and their functions as utilized directly or indirectly, actively or passively by humans to fulfil their health and well-being needs.

3.1 theoretical framework

3.1 theoretical framework

L12 disturbance factors seems to be limited to climatic event but should also include human induced interference such as over harvesting, degradation etc

We have added some human induced factors including deforestation, pollution, land degradation And wetland reclamation

I miss an explanation on what the arrows indicate. For example, the arrow between ES box and EbA approach box does not describe the same relationship as the arrow between say ecosystems and ES or between disturbance factors and ES boxes.

The arrows are simply connecting the boxes. The most important message here is that implementing EbA requires a good conceptualization of the various ecosystems, their structures, functions, threats, approaches, synergies and trade-offs. The frameworks shows that it is important to explore and understand all these aspects of ES because they are linked with the four goals which are seen as effective EbA and are important for socio-ecological resilience.

Please label all arrows. + the dotted arrows, unclear what 1. They point at (the whole or a sub circle?) and 2. What the dots indicate compared to the blue arrows.

I would strongly advise revising this visual.

Dotted arrows have been removed now.

- 🔊 Generally, the visual needs to be streamlined to be more effective, understandable at a glance.
- e.g. why the different colours for the circles? If different colours don't indicate differences in categories/types (they all seem to be underpinning factors), I would advise keeping same colours for all sub circles.
 - E.g. disturbance factor box and enhance factor box are both the same 'type/category' these are modifiers of ES quantity and quality in opposite directions, is there a simpler way of representing this?
 - Is the ecosystem box necessary? This is for a specialist audience, so might not be

We think it is necessary especially for someone who would like to implement an EbA measure. The framework can help them to visualize and brainstorm starting with the ecosystem in question (e.g. a forest ecosystem). Other adjustments to the graphic have been done to make it clearer and as suggested.

○ The strength of your study is to focus on trade offs and synergies over time and space, but the framework does not develop it that much, in my opinion, this is what the framework should be about: how to articulate and identify the trade offs and synergies across space and time of EbA so that this can support design/implementation/enforcement – possibly this could be iterative

Yes that's right. The framework helps in recognizing that even with the best EbA approaches, there are often trade-offs which need to be identified and addressed. Graphic now emphasises trade-offs and synergies more, but within a theoretical framework that also included ecosystem services and success factors for EbA approaches, which we feel are integral to this paper.

L54-55 what is systematic about this approach? Please develop + the framework does not disaggregate trade-offs and synergies. It articulates perhaps although this is not very much reflected in figure 1

We have changed this to

This framework fills this gap and proposes a holistic approach that visualizes ecosystems, their structures, functions, threats and EbA measures and associated trade-offs and synergies that need to be clearly articulated in relation to the four aspects of effective EbA.....

3.2 data

Useful table. You could add an explanation on how the different methods build and feedback on each other, at the moment they seem parallel tracks, while they certainly informed each other.

We have updated the table by showing the connection for example, conducting literature allowed us to identify the key informants and helped us to design the questions for the semi-structured interviews and focus group discussions

L14 to 23 does not add anything different to what is in the table, so it can be taken out

We have taken out those lines.

L56-57 what is a content analysis?

It is a method of analysing qualitative data such as what people have said. For example, when we analysed notes from the interviews and focus group discussions, we were looking for common words or themes that people said about threats to ecosystems or ES benefits.

L58 'to make inferences'? I find this confusing, please rephrase

That has been changed

4. background

L20-21 any indications of the share (relative to each other) and size of each of these economic activities on the Kenyan side (or even by county if available)

We have given an example of sugarcane farming in the Lake region

4.2 key ecosystem

L27-28 climate variabilities do not have high levels of risks, rephrase please.

We have changed it to 'climate variabilities affect economic activities'

L23-40 it would be useful to have an indication of the intensity of these damages. E.g. ha of forest and wetlands lost; level of pollution increase in % etc for each degradation cited.

Unfortunately, we have not managed to get any reliable empirical studies. Such information might be found in government sectoral reports (e.g. land, agriculture, Forestry) but they are notoriously hard to obtain.

4.3

L12-13 can you be more explicit on the mandate of LREB, 'facilitate policy processes' is a bit vague

The LREB was formed to drive socio-economic development in the Lake region and protect the Lake's resources through processes such as policy formulation)

L20 water resource user associations – words need to be swapped

Changed

🕒 This section needs to be refined. It should show the fragmentation and overlap across different ministries, agencies whose mandates relate to different dimensions of EbA (design, management and enforcement for example). i.e. one ministries is overseeing X which related to eba as mandated in Y legislation. Implementation is across xyz agencies which depend on different ministries.

This section should be about showing misalignment or, on the contrary, alignment between different official governance stakeholders.

That's super useful point. We have added a paragraph at the bottom of the table indicating the complexity of implementing EbA with so many players, institutions, laws, policies and strategies which are aligned and misaligned

5. synergies

On Table 3:

- 'no reported trade off' cell in the afforestation and reforestation line: there is one trade off, which is that more forest cover upstream reduces water availability downstream since forest are water users and act as water towers. Of course it means potentially improved baseflow during dry season, but it might mean decreased water availability for downstream water users

That's a possible trade-off but one which was not reported in this study. However, we have indicated a trade-off found in eucalyptus plantation which depletes groundwater resources

- 'intense competition' cell in the spring and riparian protection line: this needs to be somewhat better explained, how is the trade off of riparian protection intense competition over resource when its synergy is improved availability over resource?

This is a good observation. Only a few springs are protected which improves water availability. However, in the dry season, everyone wants to use the spring water protected due to reduced availability of water from other sources (e.g. surface flows). This has brought intense competition between different water uses and users.

Table 3, last column would be very useful as a map locating the ecosystems named. It would help text comprehension for the rest of the paper.

We have indicated on the column title that the EbA sites are located in the map (Figure 2). All the names appearing on the Table are connected with the map.

5.1 and 5.2 synergies and trade offs

This section is solely descriptive and a summary of FGDs and interviews. It also needs to be more closely referencing table 3.

I miss an understanding of who decided on the Eba measures listed. A discussion on autonomous adaptation and centrally designed and commanded adaptation would be helpful – it would tie back to section 4.3.

We have added in the last column the organizations which implemented the EbA measures. We found that the majority of the EbA measures are collaborative initiatives between the government, NGOs and the local community groups. Still the power relationships were often asymmetrical where the GOs/NGO have more power and authority because they provide the funds and technical assistance to implement the various activities.

🕒 I recommend linking 5.1 and table 3 more closely and clearly, otherwise this reads as another conceptual framework in table 3 and then summary of FGDs + a discussion is needed of who decided, who implemented and who benefits from these eba measures

We have linked 5.1 with Table 3 to show the reality of implementing EbA measures in the different sites listed in the last column

🕒 The same remark applies to 5.2, it is heavily descriptive and ideally would go into a working paper which could be referred to. + It needs to be tied to table 3

We have linked 5.2 with table 3 to show the reality of implementing EbA measures in the different sites listed in the last column

6. Governing

L53-54 the need for enabling environment and well established etc is not a conclusion drawn from the results. Or at least the description in section 5 implicit points at it but does not analyse it.

The need for enabling environment was articulated by key informants and is supported by scholarly literature. We have made changes in text along these lines.

6.1 EbA governance at national level

Please add how ecosystems and EbA are considered in the CCF mechanism, are they branded as an adaptation/mitigation opportunity? How are they included, if at all.

The CCCF supports ecosystem conservation activities such as tree planting, improved management of grazing areas or wetlands restoration . Text added accordingly.

Why is the CCF not mentioned in section 4.3? perhaps this section should be merged with the reworked 4.3.

The CCCF mechanism is now mentioned in section 4.3

Same title as 6.1? is this a mistake?

Was an error, been changed to local levels

L37-38. Low compliance is also due to weak enforcement

That's right, we have added a last sentence to reflect that despite policy formulation and reforms, weak enforcement remain a huge challenge for EbA.

7. Discussion

The start of the discussion goes over points already brought up earlier in the manuscript.

We have removed the first two sentences and maintained the fact that numerous studies on EbA have shown its importance in delivering social change

L3-4 I expected the paper to be precisely about this: who benefits directly, indirectly, who disbenefit.

We have shown that there are certainly gainers and losers with any given EbA measure, in space and time

The time dimension announced at the beginning of the paper is not prominent in the results or discussion.

A notable example with regards to time is spring protection and riparian land protection which improves water availability in the dry season but that leads to intensity of water use in protected springs where everyone goes to the springs to fetch water. We hope readers can appreciate the time dimension and that other researchers may want to explore this further in various ecosystems.

L26-28 precisely, and how does this play out in the LVB case study?

This is an important point. The same enabling environment of actors, policies etc can be disabling when power is not balanced, when policies/laws are fragmented/misaligned.

L48-51 again, and how can your articulation in this paper ensure this?

🕒 These points are all very interesting and should be the subject of the actual analysis of the paper.

We have given a reference at the bottom of section 6.1 of a participatory workshop event where stakeholders thought that there needs to be a strong political will to implement EbA for example via the CCCF.